# A HepG2 Cell-Based Biosensor That Uses Stainless Steel Electrodes for Hepatotoxin Detection

**DOI:** 10.3390/bios12030160

**Published:** 2022-03-04

**Authors:** Martin Rozman, Zala Štukovnik, Ajda Sušnik, Amirhossein Pakseresht, Matej Hočevar, Damjana Drobne, Urban Bren

**Affiliations:** 1Faculty of Chemistry and Chemical Engineering, University of Maribor, 2000 Maribor, Slovenia; martin.rozman@tnuni.sk (M.R.); zala.stukovnik1@um.si (Z.Š.); ajda.susnik@nib.si (A.S.); 2FunGlass—Center for Functional and Surface Functionalized Glass, Alexander Dubček University of Trenčín, 91150 Trenčín, Slovakia; amir.pakseresht@tnuni.sk; 3National Institute of Biology, 1000 Ljubljana, Slovenia; 4Biotechnical Faculty, University of Ljubljana, 1000 Ljubljana, Slovenia; matej.hocevar@imt.si (M.H.); damjana.drobne@bf.uni-lj.si (D.D.); 5Institute of Metals and Technology, 1000 Ljubljana, Slovenia; 6Natural Sciences and Information Technologies, Faculty of Mathematics, University of Primorska, 6000 Koper, Slovenia

**Keywords:** HepG2 cell line, impedance biosensor, adhesion, hepatotoxins, stainless steel

## Abstract

Humans are frequently exposed to environmental hepatotoxins, which can lead to liver failure. Biosensors may be the best candidate for the detection of hepatotoxins because of their high sensitivity and specificity, convenience, time-saving, low cost, and extremely low detection limit. To investigate suitability of HepG2 cells for biosensor use, different methods of adhesion on stainless steel surfaces were investigated, with three groups of experiments performed in vitro. Cytotoxicity assays, which include the resazurin assay, the neutral red assay (NR), and the Coomassie Brilliant Blue (CBB) assay, were used to determine the viability of HepG2 cells exposed to various concentrations of aflatoxin B1 (AFB1) and isoniazid (INH) in parallel. The viability of the HepG2 cells on the stainless steel surface was quantitatively and qualitatively examined with different microscopy techniques. A simple cell-based electrochemical biosensor was developed by evaluating the viability of the HepG2 cells on the stainless steel surface when exposed to various concentrations of AFB1 and INH by using electrochemical impedance spectroscopy (EIS). The results showed that HepG2 cells can adhere to the metal surface and could be used as part of the biosensor to determine simple hepatotoxic samples.

## 1. Introduction

Acute and chronic liver diseases are significant global health problems. In addition to liver damage caused by unhealthy lifestyles and viruses including hepatitis A, B, C, D, and E viruses, humans are frequently exposed to environmental hepatotoxins that can cause liver failure, acute cellular damage, or chronic diseases such as fibrosis and liver cancer [1]. Hepatotoxins include mycotoxins such as aflatoxins, which are the key concern in the food industry [2,3] and are mainly produced by *Aspergillus flavus* and *Aspergillus parasiticus* [4]. Examples of hepatotoxins include monomethylhydrazine, amatoxins, orellanine [5] and amanitin, commonly present in the fungal families of *Agaricaceae* (genus *Lepiota*) or *Amanitaceae* (genus *Amanita*). One of the most deadly mushrooms is *Amanita phalloides,* commonly referred to as the death cap, which has the highest concentration of amatoxins α-, β-, and γ-amanitin compared to other fungal families [6]. To date, no specific antidote exists for amanitin poisoning, and treatment is limited to support (dialysis, activated charcoal hemoperfusion, glucose/saline perfusion, etc.), or in the worst case, requires a liver transplant [5]. Another case of hepatotoxin represents a large variety of therapeutic drugs such as paracetamol [7], which is commonly used as pain relief medication and numerous antibiotics such as isoniazid (INH), which is often used for the treatment of more aggressive bacterial infections such as tuberculosis [8]. While INH is not directly hepatotoxic, its metabolites (e.g., acetyl hydrazine and hydrazine), along with interference with different enzymes (e.g., cytochrome P450), enables it to become harmful [9,10].

To detect harmful compounds or other analytical species, there are several methods for the detection of hepatotoxins including enzyme-linked immunosorbent assay (ELISA), high-performance liquid chromatography (HPLC), and liquid chromatography-mass spectrometry (LC-MS/MS) [11]. However, these methods require expensive equipment and are labor-intensive with regard to sample preparation [12].

One alternative is biosensors, which are small and inexpensive analytical devices that convert a biological response into an electrical signal and provide us with information about the concentration of the target analyte [13,14].

The quality of this information depends on the type of analyte components, the type of active biological component, the design of the biosensor, and the physical properties of the transducer [15]. Based on the transducer, biosensors can be classified into electrochemical, optical, and mechanical, which use an active biological component [16]. Active biological components can generally be enzymes, antibodies, in vitro cells, organelles, or tissues [17].

Many biosensors have been developed over the years that utilize cells for environmental monitoring, detection of pathological biomarkers, and cell culture monitoring [18,19,20,21,22,23,24]. Living cells have been used as biorecognition elements in biosensors since the early 1970s. They are an interesting choice of bioreceptor because they are cheaper than purified enzymes and antibodies, allow for flexibility in determining the sensing strategy, and are relatively easy and inexpensive to manufacture. However, cells can have a relatively short storage life and can be difficult to adhere to the surface [25]. Human hepatocellular carcinoma cell line (HepG2), a line derived from a human hepatoblastoma, has been found to express a variety of liver-specific metabolic functions, and is therefore a suitable candidate for biosensing applications that could mimic the liver tissue environment [26]. These cells perform many metabolic functions such as the use of the cytochrome P450 family of enzymes [27], which serve by selectively catalyzing the oxidation or reduction of chemical species as well as converting less water-soluble chemical species into smaller and more soluble metabolites [28]. As such, HepG2 cells present an excellent candidate for biosensor application, where initial chemicals are relatively harmless, but are transformed into more toxic components after their transformation.

Besides the biorecognition element, a fundamental part of a biosensor is the signal transduction mechanism through which binding events are converted into detectable and quantifiable physical signals [29]. Label-free biosensors measure integrated and phenotypic responses of whole cells with high temporal resolution. Furthermore, these biosensors enable noninvasive and highly sensitive measurements of many different cellular responses, ranging from cell adhesion, migration, differentiation, and even different types of cell death [30,31]. Optical and electrochemical biosensors are considered as label-free methods and enable real-time monitoring of the binding reaction, hence giving access to the kinetic parameters of the molecular recognition process [32,33]. In general, the factors that help in determining a particular immobilization strategy for a biosensor include the cell types used as bioreceptors, the type of material or substrate onto which cells are to be immobilized, the toxicity of the substrate, cell viability, growth requirements, and the application of the biosensor [25,34]. As a substrate surface, the most commonly used materials for biosensor development include inert metals, metal oxides, carbon-based materials (glassy carbon, graphite), and organic electroconductive polymers [35]. An interesting material represents stainless steel, since it has excellent chemical and electrical properties as well as physical and mechanical characteristics. These characteristics depend on how the steel is prepared, where it can be in different shapes such as bars, plates, or even foil. In addition to different forms, there are numerous types of steel alloys that can be used regarding the specific requirements. In the form of metal foil, this material is extremely easy to work with and has been used in a variety of electrochemical and optoelectronic devices. From an economical point, steel foil is cheaper in terms of the material cost and required tooling for its processing.

Electrochemical impedance spectroscopy (EIS) is widely used in electrochemical sensors and biosensors as a diagnostic and quantitative nondestructive detection method [36]. The measurement comprises the real (electric resistance) and imaginary (capacitance) components of the impedance response of an electrochemical system [37,38]. With electrochemical impedance spectroscopy, the frequency response of the electric current is measured, and thus the data on the living cells and the coverage of the electrode surface is obtained [39]. Under the electric fields, the cellular plasma membrane acts as an insulating barrier, directing the current to flow between or beneath the cells, leading to extracellular and transcellular currents. The extracellular current is primarily due to the intercellular conduction, while the transcellular current results from the control of cell-membrane capacitance [31]. The electrochemical impedance spectroscopy allows us to characterize the cells immobilized to the electrodes and the analyte present in the sample and due to its rapid response, it can be used to monitor molecular events in real-time [40]. Nevertheless, when designing a novel type of biosensor, different morphological or spectroscopical methods, along with cytotoxicity tests, have to be used in order to validate EIS measurements and to ensure that the device indeed has a desired response. There have been reports of biosensors that used HepG2 cells adhered to the surface [41], however, these were adhered to screen-printed carbon electrodes (SPCE), which is more difficult to prepare or modify, compared to metal electrodes.

In this paper, the authors present the use of HepG2 cells adhered to a stainless steel surface to be used as a working electrode in an electrochemical biosensor that uses the EIS method to detect different hepatotoxins such as isoniazid (INH) and aflatoxin B1 (AFB1). Cytotoxicity of both hepatotoxins was investigated, along with morphological properties of the cell-covered steel electrode and its use in the electrochemical system by the means of electrochemical impedance spectroscopy. The biosensor only uses stainless steel electrodes, greatly reducing the cost of manufacture. The data presented here, on two different chemical substances, could serve as a baseline for further improvements in order to make a biosensor that will be able to detect different varieties of hepatotoxins.

## 2. Materials and Methods

All chemicals were purchased from Sigma-Aldrich and SPI supplies, and in all cases had a purity of 98% or higher. The cells were applied to AISI 316 stainless steel (TBJ GmbH, Germany) foil. The water was purified using a Seralpur Pro 90C unit in combination with a USF Elga laboratory unit. Insulating tape (Tesa, Germany) and adsorption paper (MultiFun Paloma, Slovenia) were purchased locally. Aflatoxin B1 (AFB1) primary solution was prepared by dissolving 1 mmol of AFB1 along with 50 μL of DMSO and 4 mL of deionized water. Isoniazid (INH) primary solution was prepared by dissolving 1 mol of INH in 4 mL of deionized water. Both primary solutions were diluted with deionized water as required.

### 2.1. Cytotoxic Tests

To determine cell viability, the neutral red (NR) dye uptake test, and the Coomassie Brilliant Blue (CBB) dye test were performed in vitro on the HepG2 cell line.

The neutral red uptake assay provides a quantitative estimation of the number of viable cells in cell culture and the staining procedure was carried out according to the literature [42].

The Coomassie Brilliant Blue dye test was used as a protein determination method and performed according to the staining procedure obtained from the literature [43].

For staining, two 96-well plates with cells seeded at a density of approximately 8000 cells/hole (40,000 cells/mL) were prepared and incubated overnight at 37 °C and a 5% CO_2_ atmosphere. The cells adhered to the surface of the plate. After 24 h, aflatoxin B1 (AFB1) at concentrations of 0.1, 1, 10, 20, 50, 100, 250 μM, isoniazid (INH) at concentrations of 5, 10, 50, 100, 150, 200, 250 mM, and the negative control were prepared. The cells were exposed to the different concentrations of AFB1 and INH for 24 h. After exposure to toxins, the dyes were applied to selected individual wells and were left for a selected amount of time according to individual staining procedure before being used for further investigation. The dyed plates were investigated using a UV–Vis spectrofluorometer (BioTek, Cytation 3).

Furthermore, the resazurin test was also performed to confirm that both the neutral red and Coomassie Brilliant Blue method showed the correct results.

The reaction manifests through a visual color change from blue to a spectrum ranging from dark purple to pink [44]. The results for the resazurin dye are presented in the Appendix A along with the measurement in the figure of Section 3.5 (Appendix A).

### 2.2. Optical Microscopy

Qualitative analysis of cell viability was carried out using an optical microscope Axio (Zeiss, Germany). Cells were seeded with a density of 1.5 × 10^4^ cells/cm^2^ (2.78 × 10^4^ cells/mL) in two different 12-well plates and were incubated for 48 h at 37 °C and a 5% CO_2_ atmosphere. Cells were exposed to AFB1 at concentrations of 5, 50, and 250 μM and to INH at concentrations of 5, 50, and 150 mM for 24 h. Afterward, a qualitative exposure analysis of cell viability was carried out. All optical microscopy measurements were performed at 20× times magnification.

### 2.3. Fluorescence Microscopy

Analysis of cell viability was further carried out using a fluorescent microscope AxioImager Z1 (Zeiss, Germany). Fluorescence microscopy is a technique by which fluorescent substances are examined through a microscope. This has several advantages over other forms of microscopy, offering high sensitivity and specificity. In fluorescence microscopy, the specimen is illuminated with light of a relatively short wavelength, usually ultraviolet [45].

The 0.050 mm thick stainless steel plates with a 5 mm × 15 mm surface were first disinfected under UV light. Cells were seeded at a density of 1.5 × 10^4^ cells/cm^2^ (2.78 × 10^4^ cells/mL) on stainless steel plates and incubated for 48 h at 37 °C and a 5% CO_2_ atmosphere. Cells were exposed to AFB1 at concentrations of 5, 50, and 250 μM and to INH at concentrations of 5, 50, and 150 mM. After 24 h, the cell plates were washed with Dulbecco’s saline phosphate solution (PBS), which removed the detached cells. The cells were dyed with 2 μg/mL Hoechst 33242 and 2 μg/mL propidium iodine. Hoechst 33342 stains the nuclei of all cells blue, and propidium iodide stains the nuclei of cells with damaged plasma red. Quantitative analysis was performed using image assessment software (ImageJ 1.53k) based on the number of viable cells (blue colored) and the number of non-viable cells (red colored) on the steel surface. All fluorescence microscopy measurements were performed at 40× times magnification.

### 2.4. Scanning Electron Microscopy

After observing the adhesion of the cells on the stainless steel surface and cell viability with an optical microscope and fluorescence microscope, the surfaces with adhered cells were further prepared for scanning electron microscopy (SEM).

Surfaces with adhered cells were washed in phosphate buffer (PBS) for 30 min. Afterward, the cells on the surfaces were immersed for 48 h at 4 °C in Karnovsky fixative, prepared from 2.5% glutaraldehyde (SPI Supplies, West Chester, PA, ZDA), and 0.4% paraformaldehyde (Merck KGaA, Darmstadt, Germany) in 1 M NaH_2_PO_4_·2H_2_O and Na_2_HPO_4_·2H_2_O (Merck KGaA, Darmstadt, Germany). Cells were washed with 1 M phosphate buffer saline, fixed with 1% OsO_4_ (SPI Supplies, West Chester, PA, USA), and rinsed with distilled water. The cell surfaces were washed in an oversaturated and filtered tiocarbohydrase (TCH) solution, rinsed with distilled water, fixed with 1% OsO_4,_ and rinsed again with distilled water. Samples were dehydrated with ethanol at concentrations of 30%, 50%, 70%, 80%, 90%, and absolute ethanol (Merck KGaA, Darmstadt, Germany). Further dehydration rates were performed with a mixture of hexamethyldisiloxane (HMDS; SPI Supplies, West Chester, PA, USA) and absolute ethanol in a ratio of 1:2 and ratio of 1:1, and with the absolute HMDS. After 24 h, samples were resurfaced with a 6 nm thick coat of Au/Pd using a precision etching and coating system (PECS). Scanning electron microscopy was carried out to visualize the morphology of adhered HepG2 cells on stainless steel surfaces using a scanning electron microscope JSM-7600 F (JEOL, Tokyo, Japan). Microphotographs were recorded at 100×, 1000×, and 5000× magnification.

### 2.5. Biosensor Assembly and Electrochemical Impedance Spectroscopy (EIS)

Stainless steel plates with a surface area 5 mm × 15 mm and thickness of 0.050 mm were disinfected under UV light. Cells were seeded at a density of 3 × 10^4^ cells/cm^2^ (5.55 × 10^4^ cells/mL) on stainless steel plates and incubated for 48 h at 37 °C and 5% CO_2_ atmosphere. After 48 h, the cells were exposed to AFB1 at a concentration of 250 μM and to INH at a concentration of 150 mM. After 24 h, the plates covered with cells were washed with Dulbecco’s saline phosphate solution (PBS), which removed the detached cells. A three-electrode system with stainless steel electrodes was assembled. The device was constructed similarly to most commercial three-electrode electrochemical biosensors, utilizing flat electrodes placed in a planar configuration. The three-electrode system included the working electrode (WE) with HepG2 cells as a biological component, reference electrode (RE), and counter electrode (CE). For all three electrodes, stainless steel foil type 316 was used, with the WE being modified with HepG2 cells, while RE and CE were unmodified stainless steel foils. Electrodes were placed on a flat surface with a piece of insulation tape attached to the entire bottom of each electrode. After insulator attachment, they were positioned horizontally to each other with a 1 mm gap, in order to prevent a short circuit. A piece of adsorption paper with dimensions of 5 mm × 17 mm was placed on top of the electrodes in such way that each electrode had an active area of 5 mm × 5 mm. A phosphate buffer solution (PBS) was used as the electrolyte and was carefully dripped onto adsorption paper until it was soaked.

For the evaluation of electrochemical cells, the electrochemical impedance spectroscopy method (EIS) was used [46]. The measurements were carried out with a PalmSens4 potentiostat/galvanostat with a frequency range from 50 kHz to 0.1 Hz with 9.5 points per decade, totaling 55 points per measurement. The stability of the three-electrode system with HepG2 cells was previously determined by measuring the potential of the open circuit (OCP), where the potential difference between the reference electrode (RE) and the working electrode (WE) was measured.

## 3. Results

### 3.1. Cytotoxic Tests

The neutral red assay showed that AFB1 causes statistically significant changes in the lysosomal integrity of HepG2 cells at concentrations above 10 μM and that the INH causes changes in lysosomal integrity at concentrations above 10 mM and the changes can be observed in Figure 1A,B, where the decrease in the cell viability with increasing hepatotoxin concentration is shown. The Coomassie Brilliant Blue assay showed similar results (Figure 1C,D) compared to the neutral red assay, indicating that both methods imply that both compounds indeed have a hepatotoxic response. Similarly to data presented in the neutral red test, AFB1 concentrations above 10 μM showed a reduction in the number of cells in the cell population and that INH concentrations above 10 mM also had a significant effect on the number of cells.

Overall, the cytotoxic assays showed that AFB1 had significant effects on cell proliferation and viability at concentrations exceeding 10 μM, while INH showed effects at concentrations above 10 mM. The results were used to determine what concentration range is expected for an individual substance to have an effect and to assess the limit of detection (LOD) for the investigated biosensor.

### 3.2. Optical Microscopy

The surface of the glass plate had good coverage with the HepG2 cell population. In comparison, HepG2 cells had progressively lower cell coverage on the plate after exposure to increasing concentrations of AFB1 and increasing concentrations of INH. Although all hepatocytes appeared to be homogeneous under the light microscope, it can be seen that the cells in the different zones showed structural and functional heterogeneity. The photomicrographies for both hepatotoxins are presented in Figure 2. For AFB1 exposure (Figure 2A–D), it can be observed that the gaps between the cells increased, which indicates that their affinity toward tissue structure tendency was reduced. At a concentration of 50 µM, AFB1 cells began to individualize, suggesting that they were less able to proliferate, with an increasing number of smaller, cell-like structures, suggesting that some of the cells had started the cell death process [47]. At the highest concentration (250 µM AFB1), the only visible structures were blebs and cell residues that appear during the cell death process. Similar damage could be observed in the case of INH (Figure 2E–H), where at the initial concentration (5 mM INH), cell proliferation was again decreased, and at higher concentrations (50 mM and 250 mM), smaller structures could be observed, which could be due to cell necrosis caused by the increased INH concentration.

### 3.3. Fluorescence Microscopy

Using fluorescence microscopy, cell survival was quantitatively determined by counting all cells in the cell population (blue cell nucleus) and dead cells (red cell nucleus). Two different staining agents were used for this purpose: Hoechst 33342 stains the nucleus of all cells blue, and propidium iodide stains the nucleus of cells with damaged plasma red. The difference between the number of all cells and the dead cells in the cell population was calculated to obtain the number of all living cells in the cell population (the difference between blue and red nuclei). Fluorescence photomicrographs are presented in the Appendix A. For both AFB1 and INH, the results showed that the HepG2 control had a high concentration of adherent cells on the steel surface throughout the cell population. The number of live cells on the steel surface was relatively high compared to the total cell population, while the number of dead cells on the stainless steel surface was quite low when observing cells in the control group. The number of all cells in the whole cell population and the number of live HepG2 cells on the steel surface after exposure to increasing AFB1 concentrations of 5, 50, and 250 μM decreased, consequently, the number of dead cells at increasing concentrations of AFB1 gradually increased compared to the control. In comparison, HepG2 cells showed progressively lower cellular steel coverage after exposure to increasing concentrations of AFB1, which was expected due to its hepatotoxicity. For INH investigation, the number of dead cells gradually increased at concentrations of 5 and 50 mM compared to the control. At an INH concentration of 250 mM, dead cells on the stainless steel surface appeared to be lower than at an INH concentration of 50 mM, which could be due to detachment of the cells or their complete necrosis due to the high concentration of INH. Graph of counted cells during fluorescence microscopy is presented in Figure 3.

### 3.4. Fluorescence Microscopy

The effect of AFB1 and INH on the morphology of HepG2 cells was studied by scanning electron microscopy (SEM). HepG2 cells were exposed to different concentrations of AFB1 (5, 50, and 250 μM) and different concentrations of INH (5, 50 in 150 mM) for 48 h. The results are presented separately for AFB1 and INH. Microphotographs at different magnifications are presented in the Appendix A.

#### 3.4.1. Aflatoxin B1

The results indicate that AFB1 at increasing concentrations induces intracellular biochemical events such as skeletal damage, cell growth inhibition, and cell death. Therefore, the results showed distinct morphological changes and decreased cell viability and adhesion to the stainless steel surface with increasing AFB1 concentration. The contrast in the photographs was due to the contrast reagent used in cell fixation. The elongated lines that can be observed in the background of the cells are an example of cold-rolled steel.

Figure 4 shows thee SEM microphotographs of HepG2 cells at various concentrations of AFB1 (0, 5, 50, and 250 μM) at 1000× magnification. During morphology investigation, it was observed that the cell surface was composed of flattened cells that interacted closely. At an AFB1 concentration of 5 μM, a rough and textured cell surface was observed, where cells were more widely spaced and elevated. At an AFB1 concentration of 50 μM, the cell surface was more clearly altered, roughened, and textured, with the cells becoming flatter and had a spindle or round shape with growths up to 1 μm in diameter. At an AFB1 concentration of 250 μM, an even more structured surface with an altered round cell shape was observed, which could indicate that the cell has gone well into smaller structures, described as blebs. These can be identified as the spherical structures observed on the surface of the electrode.

#### 3.4.2. Isoniazid

The results show that INH induces intracellular biochemical events such as skeletal damage, cell growth inhibition, and cell death, which can be due to either apoptosis and necrosis, at increasing concentrations. The SEM microphotographs (Figure 5A–D) demonstrate distinct morphological changes such as reduced adhesion, followed by cell disintegration with increasing INH concentration. Similar to the SEM analysis of samples exposed to AFB1, the elongated lines that were observed in the background of the cells are an example of the cold-rolled steel surface.

Figure 5 shows representative SEM figures of HepG2 cells at various concentrations of INH (0, 5, 50, and 250 mM) at 1000× magnification. Again, in the control group, the cells were flattened and interacted closely with each other. At an INH concentration of 5 mM, the cell surface was smooth, more spaced, and stretched, suggesting that their behavior was altered. At 50 mM INH, the HepG2 cells had a noticeable rounder shape with individual vesicular growths of up to 1 μm. At even increased concentrations of 250 mM, the cell surface was transformed into several individual round clusters, with even more significant vesicular proliferations. Based on the fluorescence microscopy figures, it can be assumed that these spherical cell residues are blebs that formed during the cell death process, which can also be observed as red spots stained with propidine iodide. Despite the cell death, the cell structures appeared to not fully detach and a residue remained on the steel surface.

### 3.5. Electrochemical Impedance Spectroscopy

Electrochemical impedance spectroscopy (EIS) measurements were performed to determine the usefulness of adherent HepG2 cells on a stainless steel surface as a biosensing electrochemical device. Both of the Nyquist plots are presented in Figure 6. In all of the EIS figures, the black label represents the negative control where the cells were not affected by hepatotoxin, blue label represents the biosensor with added hepatotoxin, while the red label represents the positive control, where no cells were on the surface. The results show that the negative control, where the cells were on the surface but not exposed to the toxin, had a relatively low total impedance from 120 to 160 kΩ cm^−2^ at low frequencies (below 10 Hz). This difference in impedance can be attributed to cell coverage, where the electrode could have exposed surfaces, resulting in lower impedance. It can be concluded that the coating of the cells on the surface was relatively thin or that the cells did not completely cover the electrode [48]. Despite this, an impedance response could still be observed, since there was a change in exposed metal surface, which affects the general resistance and conductivity of the active surface area. The Nyquist plot showed that the capacitance (-Z’) of the negative control was higher than the capacitance of both the investigated samples and the positive control, suggesting that the surface conductivity increased with the addition of hepatotoxin. For AFB1, a test concentration of 250 µM was used, while for INH, a test concentration of 250 mM was used.

In the case of AFB1 testing (Figure 6A), it can be observed that the tested concentration almost aligned with the positive control, suggesting that the surface had been almost completely stripped from the cells. This slightly higher impedance could be attributed to the cell residue and blebs [49] that remain on the surface and can be observed during fluorescence microscopy and SEM studies. From the Bode plot (Figure 6C), it can be observed that overall impedance |Z| had a relatively small change for the duration of measurement, while comparing phase change in frequency range below 1 Hz showed significant differences between the untreated negative control and the treated biosensor. In the case of the treated sample, the response was very similar to the positive control, which had no cells attached onto WE.

In the case of INH testing (Figure 6B), the difference between the negative and positive control was smaller, with a difference between 10 and 20 kΩ cm^−2^, which is likely due to the lower coverage of HepG1 cells on the steel surface. Furthermore, when measuring the response for INH, this actually had a slightly lower impedance compared to the positive control. The most likely reason for this is the increased conductivity, which could be due to secondary metabolites, or products that are formed during the necrosis process of the cells [50]. Similar to AFB1 testing, the Bode plot (Figure 6D) showed that overall impedance |Z| remained relatively unchanged between all of the measurements, indicating only a slight change when comparing the negative control, positive control, and sample. Difference in phase angle, however, could be observed, and was visible at frequencies below 1 Hz, indicating a change in the electrode surface.

Finally, the obtained results were modeled using an R [CQ] fitting circuit (Figure 6E), which shows that the electrochemical behavior can be interpreted to the following electrical elements. Despite certain differences, it is expected that the initial resistor *R* covers the combined effects of the electrolyte and steel surface covered with cells, while the constant phase element (CPE) *Q* can be attributed toward cell coverage, where capacitor *C* determines the use of cellulose paper, which acts as a electrolyte holder material. Previous studies have used RQ [51,52] and R [RQ] [53] circuits for fitting, which was also investigated, however, it was observed during the fitting process that the data were better fitted if the parallel resistor was replaced with a capacitor. The tables of fitting results are shown in Appendix A.

Schematics of the biosensor are shown in Figure 6E, indicating the electrode placement.

The results are consistent with other microscopic observations confirming that both AFB1 and INH damages the HepG2 cells at high concentrations. The data are in agreement with the fitting circuit and previous literature [54], suggesting that the change in impedance is indeed due to changes on the electrode surface [55]. The fitting circuit and spectra are presented in the Appendix A. It is speculated that due to the relatively uneven coverage and cell gaps on the electrodes, the limit of detection (LOD) is predicted to be quite high for AFB1 and 250 mM for INH, since it was difficult to obtain a signal with a sufficient difference at lower concentrations. In addition, the data showed that it is possible to detect hepatotoxins even with electrodes that are not fully covered with cells, since the response detection is dictated by surface resistance and conductivity change. Therefore, the collected data indicate that it is possible to detect the presence of different types of hepatotoxins, although at relatively high concentrations.

## 4. Discussion

The cytotoxicity effects of AFB1 and INH on HepG2 cells were investigated along with the ability of HepG2 cells to adhere to the stainless steel surface and the use of the cell covered stainless steel surface in order to be used as an electrochemical biosensor. The HepG2 cells reacted to hepatotoxin aflatoxin B1 (AFB1) and isoniazid (INH) regardless of the adhered surface (glass or steel). In cytotoxic assays, AFB1, at a concentration above 10 μM, and INH at a concentration above 10 mM, were found to significantly reduce the viability of HepG2 cells. These results were further confirmed using fluorescence microscopy. The number of all cells in the total cell population and the number of live HepG2 cells on the steel surface after exposure to increasing concentrations of AFB1 (0, 5, 50, and 250 μM) and INH (5, 50, and 150 mM) decreased, while the number of dead cells gradually increased with increasing concentrations of AFB1 (0, 5, 50, and 250 μM) and INH (0, 5, 50, and 150 mM). With a fluorescence microscope, it was observed that the cell population control had relatively good coverage of the steel surface, while HepG2 cells gradually decreased after exposure to increasing concentrations of the toxins AFB1 and INH. With optical microscopy, it was also observed that the cells used in the negative control sample achieved good statistical coverage of the plate surface. This indicates that the concept of adhering cells to the metal surface is plausible and enables its initial use as an electrode material while HepG2 cells showed decreasing coverage of the plate after being exposed to increasing INH and AFB1 concentrations. Using scanning electron microscopy (SEM), it was observed that the toxin affects the morphology, adhesion, and coverage with adhered HepG2 cells of metal electrodes. At increasing concentrations, AFB1 and INH cause intracellular biochemical events such as skeletal damage, cell growth inhibition, and cell death. Therefore, the SEM figures showed distinct morphological changes and a decrease in cell viability and adhesion to the steel surface with increasing concentrations of the toxins. The electrochemical impedance spectroscopy results are consistent with other microscopic observations, confirming that INH at high concentrations damages HepG2 cells, and consequently, they no longer cover the metal surface. From the data collected, it can be concluded that a biosensor using live HepG2 cells could successfully detect the presence of INH at a relatively high concentration of 150 mM as well as for AFB1 at concentrations exceeding 250 µM. Overall, the device demonstrated the ability to determine if it was exposed to hepatotoxins.

## 5. Conclusions

All of the obtained results support the hypothesis that HepG2 cells can adhere to the metal surface. It was shown that the prepared metal electrodes with adhered cells could be used as part of the biosensor to determine simple hepatotoxic samples. Despite the fact that these concentrations were substantially higher that the ones that showed effects on cell viability in the cytotoxicity test, the cell-covered electrode successfully showed that it could detect both AFB1 and INH. In addition, such biosensors could be, with minimal modifications, used to detect other hepatotoxins such as mycotoxins and could be used as an effective tool in food industries or forensics. In addition, the results suggest that in the future, it may be possible to use metal electrodes with attached living cells for practical biosensors that could be generally useful in medical diagnostics or environmental monitoring. An impedance biosensor could be used for differžent cell types (lung, nasal, skin, nerve cells, and others) that could probe different types of toxins and pathogens specific to the type of target disease or suspected toxin in the environment.

## Figures and Tables

**Figure 1 biosensors-12-00160-f001:**
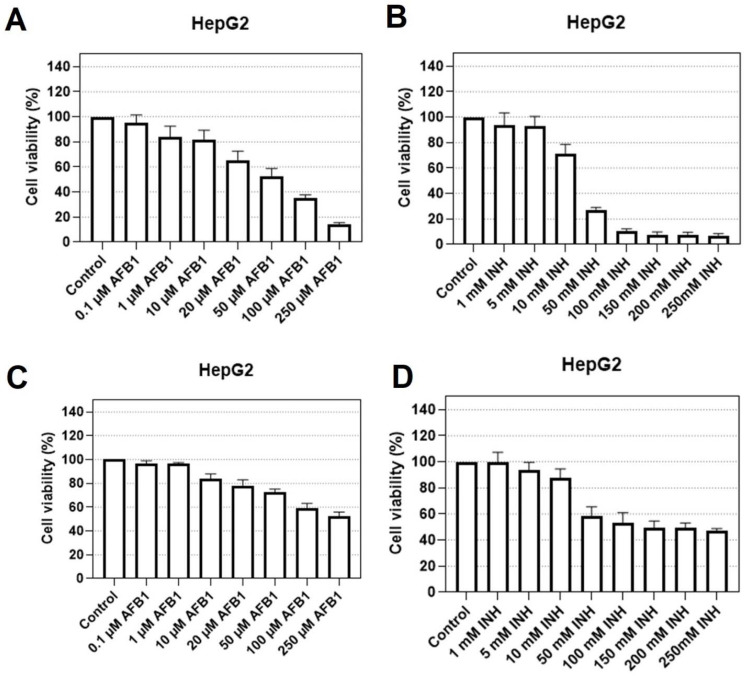
The HepG2 cell viability after exposure to various concentrations of AFB1 (**A**) and INH (**B**) when stained with neutral red and cell viability following the Coomassie Brilliant Blue staining test for AFB1 (**C**) and INH (**D**). The results are given as mean values of the three measurements ± STD (*p* < 0.001).

**Figure 2 biosensors-12-00160-f002:**
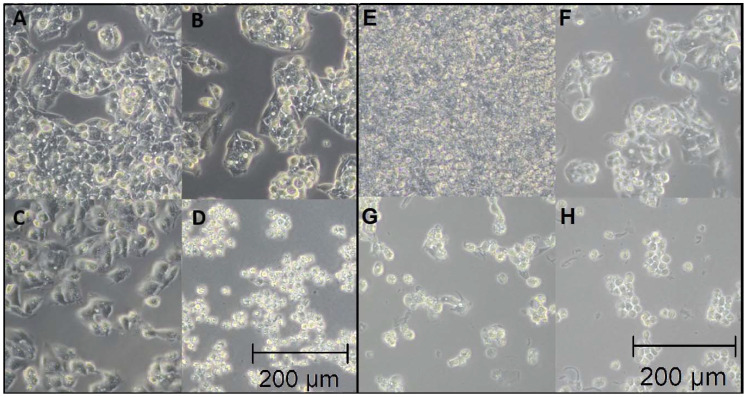
Optical microscopy photographs under the influence of AFB1 on the HepG2 cell line (**A**–**D**) exposed to different concentrations for the control (**A**), 5 µM (**B**), 50 µM (**C**), and 250 µM (**D**) AFB1 concentrations. A similar investigation was conducted for INH (**E–G**) showing the control (**E**), 5 mM (**F**), 50 mM (**G**), and 250 mM (**H**) INH concentration. For all photographs, magnification was 20×.

**Figure 3 biosensors-12-00160-f003:**
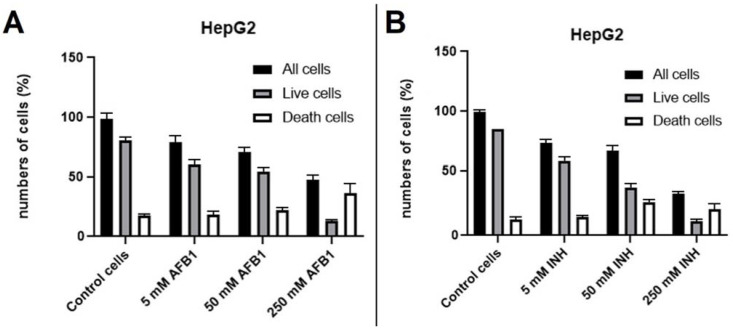
Impact of AFB1 and INH on HepG2 cell survival; graphs of the fluorescent microscopy of HepG2 cells exposed to AFB1 (0, 5, 50, and 250 μM) (**A**) and INH (0, 5, 50, and 250 mM) (**B**) at 24 h using the double staining method with Hoechst 33342 and propidine iodide (PI).

**Figure 4 biosensors-12-00160-f004:**
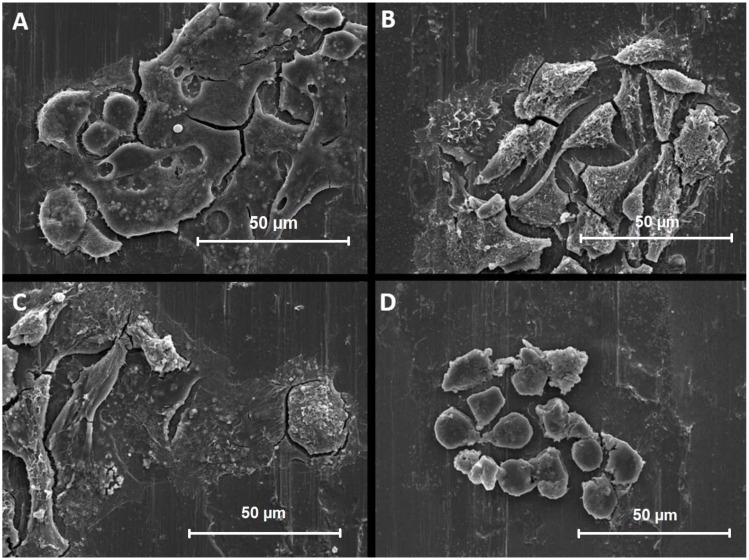
SEM photographs of the HepG2 cells exposed to increasing concentrations of AFB1 at 1000× magnification. HepG2 cells were exposed to different concentrations starting with the control (**A**), 5 µM (**B**), 50 µM (**C**), and 250 µM (**D**).

**Figure 5 biosensors-12-00160-f005:**
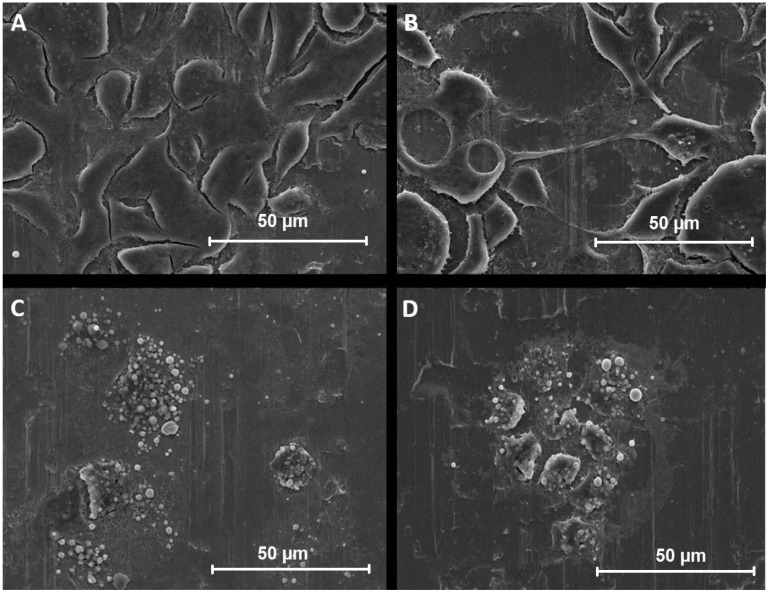
SEM photographs of the HepG2 cells exposed to increasing concentrations of INH at 1000× magnification. HepG2 cells were exposed to different concentrations starting with the control (**A**), 5 mM (**B**), 50 mM (**C**), and 250 mM (**D**).

**Figure 6 biosensors-12-00160-f006:**
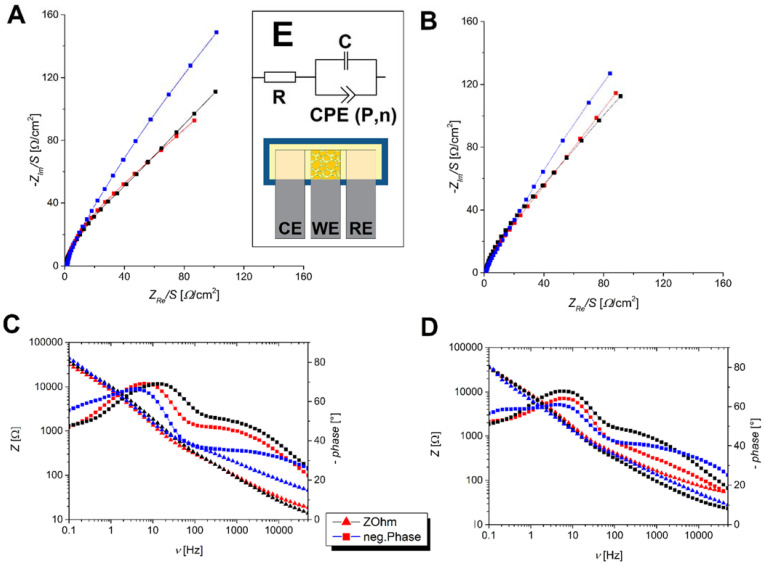
Nyquist diagrams of investigated biosensors showing a response for aflatoxin B1 (**A**) and isoniazid (**B**), along with Bode diagrams for aflatoxin B1 (**C**) and isoniazid (**D**). For Bode diagrams, triangles represent total impedance |Z|, while squares represent phase change. Both biosensors that had applied hepatotoxin are marked with a black label, where the control measurement is marked with the blue label and the biosensor without any attached cells is marked with the red label. Units are Ω/cm^2^ where the impedance response is normalized for the surface of the WE. A schematic of the biosensor with electrode placement (**E** bottom) along with fitted circuit for EIS analysis (**E** top) is also shown.

## Data Availability

Data are available directly from the authors upon reasonable request.

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
