# Peer review of "A HepG2 Cell-Based Biosensor That Uses Stainless Steel Electrodes for Hepatotoxin Detection"

_biosensors, 2022, doi:10.3390/bios12030160_

Round 1

Reviewer 1 Report

Ms title: A HepG2 cell-based biosensor that uses stainless steel electrodes for hepatotoxin detection

Journal: Biosensors (MDPI)

Comments:

The authors presented a cell-based biosensor for hepatotoxin detection with various optical and electrochemical methods. The manuscript is nicely written, clear, and to the point. However, there are some revisions are needed to publish this manuscript in the journal. The specific comments are mentioned below which authors are encouraged to address in their revised manuscript.

Specific comments:

  1. Page 5, line 223. “AFB1 causes….. Concentrations above 1µM and that the INH causes changes in lysosomal integrity at concentrations above 10 mM” I can see significant change for INH 10 mM, however AFB1 at 1µM didn’t exhibit any significant change, rather it is linearly decreasing with increasing concentration. Authors are encouraged to address this in their revised manuscript.
  2. Page 11, line 359. Relatively low total impedance of up from 120 to 160 Kohm/cm^2 at low frequencies (below 10 Hz)… It is difficult to assess the resistance values from the given Nyquist plots in Figure 6. Authors are encouraged to append Bode plots to Figure 6, and replace the Nyquist plots (Z,’ Z’’) with complex capacitance plots (C’, C’’).
  3. Line 362, It can be concluded that the coating of the cells on the surface is relatively thin or that the cells do not completely cover the electrode. The authors need to explain this in detail.
  4. The authors also encourage to fit the impedance spectroscopy results to the R[RQ] model and compare the results with the present R[CQ] model.
  5. Line 386. The date is in agreement with the fitting circuit and previous literature. The authors need to add the fitting circuit in the manuscript and discuss it in detail which is in agreement with previous literature.

Further, there are some minor corrections are expected in the revised manuscript, thus I recommend minor revision on this manuscript.

Author Response

Reviewer 1:

Comments:

The authors presented a cell-based biosensor for hepatotoxin detection with various optical and electrochemical methods. The manuscript is nicely written, clear, and to the point. However, there are some revisions are needed to publish this manuscript in the journal. The specific comments are mentioned below which authors are encouraged to address in their revised manuscript.

 Specific comments:

  1. Page 5, line 223. “AFB1 causes….. Concentrations above 1µM and that the INH causes changes in lysosomal integrity at concentrations above 10 mM” I can see significant change for INH 10 mM, however AFB1 at 1µM didn’t exhibit any significant change, rather it is linearly decreasing with increasing concentration. Authors are encouraged to address this in their revised manuscript.

There was a typo in the text where it was intended to be 10 µM instead of 1 µM. It was corrected over the course of entire manuscript.

  1. Page 11, line 359. Relatively low total impedance of up from 120 to 160 Kohm/cm^2 at low frequencies (below 10 Hz)… It is difficult to assess the resistance values from the given Nyquist plots in Figure 6. Authors are encouraged to append Bode plots to Figure 6, and replace the Nyquist plots (Z,’ Z’’) with complex capacitance plots (C’, C’’).

Bode plots (Line 439) have been prepared and are attached in the main text as advised by the reviewer, however, Nyquist plots were kept in the figure, since they are more common and are more easily interpreted by general electrochemistry and material science audience.

  1. Line 362, It can be concluded that the coating of the cells on the surface is relatively thin or that the cells do not completely cover the electrode. The authors need to explain this in detail.

The coatings have been made using modified procedure. As the cover process is not complete, the EIS measurements rely on difference in free surface exposure, that changes, when cells are exposed to toxin and start to disintegrate (or de-attach). Sentences have been added (line 455-458) that explain this in text.

  1. The authors also encourage to fit the impedance spectroscopy results to the R[RQ] model and compare the results with the present R[CQ] model.

The authors have investigated the R[RQ] model as well. While the Nyquist plot and absolute impedance vs. frequency (|Z| vs. freq.) are very similar, the difference can be observed in the phase shift, which is closer to the present R[CQ] model. Both phase shifts are in certain parts misaligned, however, since R[CQ] model better describes phase shift in low frequency region (phase region associated with surface changes) we decided to incorporate this model. To support this claim, the draft of both phase shift simulations is present as a separate figure in this response. This issue has also been incorporated into the paper in the part 3.5 Electrochemical impedance spectroscopy (line 427-435).

Figure 1: Comparison between R[CQ] and R[RQ].

  1. Line 386. The date is in agreement with the fitting circuit and previous literature. The authors need to add the fitting circuit in the manuscript and discuss it in detail which is in agreement with previous literature.

The fitting circuit has been added to the Figure 6 (Line 439-447) as the reviewer suggested, along with the schematics of the biosensor and detailed explanation regarding fitting parameters (line 427-438).

Further, there are some minor corrections are expected in the revised manuscript, thus I recommend minor revision on this manuscript.

The paper has been revised in details for any other grammatical errors, which were corrected.

Reviewer 2 Report

In this paper, the authors made a HepG2 cell-based biosensor that used stainless steel electrodes to detect hepatotoxins (INH, AFB1).

The manuscript is well-organized and fairly satisfied. But there are some issues to be addressed for the publication:

  1. English should be corrected carefully to remove grammatical errors in the manuscript.
  2. In introduction section, Line 70: The introduction of HepG2 cells is too short to indicate the hepatotoxin biomarker pathology. Please make this issue clear in the introduction part. 
  3. There is no introduction to stainless steel electrodes in the introduction section. Please add its introduction to the manuscript.
  4. Please change cells/cm2 to cells/cm2 in the paper.
  5. In Section 2.5, The authors used the stainless steel plate as a working electrode. What are the materials for reference and counter electrodes? Is it the same stainless steel plate used for making reference and counter electrodes?
  6. There is no schematic showing the configuration of the electrode and the construction of the HepG2 cell-based biosensor. Please add a scheme to point this out.
  7. In Figure 3: There is no error bar (or standard deviation) in each measurement. Please add them.
  8. What is the S in Figure 6? Is it the surface area? Please explain this.
  9. For monitoring the cell-impedance, the Bode plot is often used instead of the Nyquist plot. It is better that the authors insert the Bode plots into the paper.

Author Response

Reviewer 2:

In this paper, the authors made a HepG2 cell-based biosensor that used stainless steel electrodes to detect hepatotoxins (INH, AFB1).

The manuscript is well-organized and fairly satisfied. But there are some issues to be addressed for the publication:

  1. English should be corrected carefully to remove grammatical errors in the manuscript.

The manuscript was carefully inspected for any grammatical errors.

  1. In introduction section, Line 70: The introduction of HepG2 cells is too short to indicate the hepatotoxin biomarker pathology. Please make this issue clear in the introduction part. 

The hepatotoxin biomarker pathology has been updated, covering HepG2 performance regarding different metabolic functions and ability to convert initial chemical species into different metabolites (line 80-85).

  1. There is no introduction to stainless steel electrodes in the introduction section. Please add its introduction to the manuscript.

An explanation for qualities of stainless steel foil has been incorporated into introduction, along with the references that support these claims (line 101-108).

  1. Please change cells/cm2 to cells/cm2 in the paper.

The text was corrected to appropriate units (cells/cm2) over the entire text.

  1. In Section 2.5, The authors used the stainless steel plate as a working electrode. What are the materials for reference and counter electrodes? Is it the same stainless steel plate used for making reference and counter electrodes?

All three electrodes used stainless steel, with working electrode using steel foil coated with cells, while the other two electrodes (reference and counter) used unmodified steel foil. An explanation regarding electrode material was incorporated into the Section 2.5 (line 236-238).

  1. There is no schematic showing the configuration of the electrode and the construction of the HepG2 cell-based biosensor. Please add a scheme to point this out.

The schematic showing the biosensor has been added in the figure 6 (line 439).

  1. In Figure 3: There is no error bar (or standard deviation) in each measurement. Please add them.

The error bar has been added in Figure 3 (line 324).

  1. What is the S in Figure 6? Is it the surface area? Please explain this

The S stand for surface area normalized for entire surface area of working electrode. The explanation has been included in the text (line 445)

  1. For monitoring the cell-impedance, the Bode plot is often used instead of the Nyquist plot. It is better that the authors insert the Bode plots into the paper.

As per reviewers request, the Bode plots have been integrated into Figure 6, along with its interpretation in section 3.5 (lines: 390-393; 399-401; 410-415; 422 -437)

Reviewer 3 Report

The present manuscript reported A HepG2 cell-based biosensor that uses stainless steel electrodes for hepatotoxin detection. The paper can be interesting to biosensing applications in the new area. Therefore, I can recommend it for publication after improvement!

---- At the first, please explain the novelty and also advantage(s) of your work, and then, please compare the purposed methodology with reported methods.

----INH is a recognized antibiotic used for the treatment of tuberculosis, please give more information to the readers about it, and explain how it can induce cell damage? Directly or indirectly?

----Why did you choose “stainless steel electrodes “?

----Fig.6, please add more information about the curves and colors, it is not clear. Please check the others figures.

----Page9.  Isoniazid, you explained why INH can cause the hepatotoxic conditions. But You should provide suitable references.

----Could you explain, why you use two methods: the neutral red assay (NR), and the Coomassie brilliant blue (CBB) assay.  

----- Figure 5, could you show the cells in the image? The SEM images are not clear for the readers.

Please add these references related to cell monitoring.

  1. Wang, N., Wang, H., Zhang, J., Ji, X., Su, H., Liu, J., ... & Zhao, W. (2022). Diketopyrrolopyrrole-based sensor for over-expressed peroxynitrite in drug-induced hepatotoxicity via ratiometric fluorescence imaging. Sensors and Actuators B: Chemical, 352, 130992.
  2. Lu, Wei, Bingtong Huang, Yongcheng He, Jiao Yang, and Yingchun Li. "A facile cell-involved microfluidic platform for assessing risk of hepatotoxic chemicals via on-line monitoring of multi-indexes." Sensors and Actuators B: Chemical 341 (2021): 129938.
  3. Zhang, Xinwei, Amir Hatamie, and Andrew G. Ewing. "Nanoelectrochemical analysis inside a single living cell." Current Opinion in Electrochemistry 22 (2020): 94-101.
  4. Zhou, Wenli, Karen Graham, Baltasar Lucendo-Villarin, Oliver Flint, David C. Hay, and Pierre Bagnaninchi. "Combining stem cell-derived hepatocytes with impedance sensing to better predict human drug toxicity." Expert Opinion on Drug Metabolism & Toxicology 15, no. 1 (2019): 77-83.

Author Response

Reviewer 3:

Comments and Suggestions for Authors

The present manuscript reported A HepG2 cell-based biosensor that uses stainless steel electrodes for hepatotoxin detection. The paper can be interesting to biosensing applications in the new area. Therefore, I can recommend it for publication after improvement!

---- At the first, please explain the novelty and also advantage(s) of your work, and then, please compare the purposed methodology with reported methods.

The novelty and advantage have been explained in introduction (Line 122-127 and 133-136) clearly stating that the stainless steel is more robust and cost-effective material compared to screen printed carbon. In addition, methodology has been explained, showing importance of combined morphological, spectroscopical and electrochemical investigations. The novelty has also been described in in the introduction (lines 133-136), along with improved methodology.

----INH is a recognized antibiotic used for the treatment of tuberculosis, please give more information to the readers about it, and explain how it can induce cell damage? Directly or indirectly?

The mode of action for INH, has been explained in introduction explaining, that INH o its own is not directly hepatotoxic, but can induce cell damage via metabolites (hydrazine) and interference with Cytochrome P450 family of enzymes (line 49-50).

----Why did you choose “stainless steel electrodes “?

Stainless steel electrodes were chosen for their excellent mechanical and chemical properties, as well as low cost. This has been now explained in the lines 101-109 and 133-136.

----Fig.6, please add more information about the curves and colors, it is not clear. Please check the others figures.

The figure 6 (line 439) has been corrected with more detailed description.

----Page9.  Isoniazid, you explained why INH can cause the hepatotoxic conditions. But You should provide suitable references.

References regarding Isoniazid hepatotoxicity along with suitable references has been added in Introduction (line 49-55)

----Could you explain, why you use two methods: the neutral red assay (NR), and the Coomassie brilliant blue (CBB) assay.  

Both NR and CBB assay were used in order to assure the accuracy of the results and to exclude the any potential contaminants that could be appear when preparing the chemicals used in assay.

----- Figure 5, could you show the cells in the image? The SEM images are not clear for the readers.

The SEM images represented in Figures 4 and 5 show gradual change of adhered structures that are attached on the stainless steel surface. The cells themselves in vivo, can be observed in Figure 2, while the SEM images were intentionally left at 1000 magnification in order to show cell residue that is left on the surface after interaction with hepatotoxin. Considering this, additional figures showing SEM images at lower (100x) and higher (5000x) magnification are presented in supplementary information.

Please add these references related to cell monitoring.

  1. Wang, N., Wang, H., Zhang, J., Ji, X., Su, H., Liu, J., ... & Zhao, W. (2022). Diketopyrrolopyrrole-based sensor for over-expressed peroxynitrite in drug-induced hepatotoxicity via ratiometric fluorescence imaging. Sensors and Actuators B: Chemical, 352, 130992.
  2. Lu, Wei, Bingtong Huang, Yongcheng He, Jiao Yang, and Yingchun Li. "A facile cell-involved microfluidic platform for assessing risk of hepatotoxic chemicals via on-line monitoring of multi-indexes." Sensors and Actuators B: Chemical 341 (2021): 129938.
  3. Zhang, Xinwei, Amir Hatamie, and Andrew G. Ewing. "Nanoelectrochemical analysis inside a single living cell." Current Opinion in Electrochemistry 22 (2020): 94-101.
  4. Zhou, Wenli, Karen Graham, Baltasar Lucendo-Villarin, Oliver Flint, David C. Hay, and Pierre Bagnaninchi. "Combining stem cell-derived hepatocytes with impedance sensing to better predict human drug toxicity." Expert Opinion on Drug Metabolism & Toxicology 15, no. 1 (2019): 77-83.

As per reviewers suggestion, the references have been incorporated into introduction part covering cell monitoring. (line 71)
